# Estimated Burden of Fungal Infections in Oman

**DOI:** 10.3390/jof7010005

**Published:** 2020-12-23

**Authors:** Abdullah M. S. Al-Hatmi, Mohammed A. Al-Shuhoumi, David W. Denning

**Affiliations:** 1Department of microbiology, Natural & Medical Sciences Research Center, University of Nizwa, Nizwa 616, Oman; 2Department of microbiology, Centre of Expertise in Mycology Radboudumc/CWZ, 6500 Nijmegen, The Netherlands; 3Foundation of Atlas of Clinical Fungi, 1214GP Hilversum, The Netherlands; 4Ibri Hospital, Ministry of Health, Ibri 115, Oman; mls971.ihs@gmail.com; 5Manchester Fungal Infection Group, Manchester Academic Health Science Centre, The University of Manchester, Manchester M13 9PL, UK; ddenning@manchester.ac.uk

**Keywords:** fungal diseases, superficial, invasive, epidemiology, incidence, prevalence, Oman

## Abstract

For many years, fungi have emerged as significant and frequent opportunistic pathogens and nosocomial infections in many different populations at risk. Fungal infections include disease that varies from superficial to disseminated infections which are often fatal. No fungal disease is reportable in Oman. Many cases are admitted with underlying pathology, and fungal infection is often not documented. The burden of fungal infections in Oman is still unknown. Using disease frequencies from heterogeneous and robust data sources, we provide an estimation of the incidence and prevalence of Oman’s fungal diseases. An estimated 79,520 people in Oman are affected by a serious fungal infection each year, 1.7% of the population, not including fungal skin infections, chronic fungal rhinosinusitis or otitis externa. These figures are dominated by vaginal candidiasis, followed by allergic respiratory disease (fungal asthma). An estimated 244 patients develop invasive aspergillosis and at least 230 candidemia annually (5.4 and 5.0 per 100,000). Only culture and microscopy are currently available for diagnosis, so case detection is suboptimal. Uncertainty surrounds these figures that trigger the need for urgent local epidemiological studies with more sensitive diagnostics.

## 1. Introduction

Oman is located in the Middle East on the Arabian Peninsula and has an area of 309,500 square kilometers. It borders the Arabian Sea, Oman Sea, Arabian Gulf, also known as the Persian Gulf and shares a border with the United Arab Emirates, the Republic of Yemen, and the Kingdom of Saudi Arabia. It has a desert climate, which leads to hot, dry conditions in the interior and a hot, humid climate along the coast. According to the National Center for Statistics and Information (NCSI) estimation, Oman has a total population of about 2,579,236 in 2018. The Omani population shows a sex ratio of 102 males per 100 females [1].

The health services in the Sultanate of Oman have developed tremendously over the past years. Oman’s healthcare system has taken serious efforts since the 1970s to improve its quality management services [2]. In 2000, the World Health Organization (WHO) ranked Oman’s healthcare system as one of the best 10 healthcare systems in the world [3]. However, in Oman, there is a major lack of information on the burden of fungal disease. The clinical diagnosis of severe fungal infection is often impossible but always challenging, and recognition of infection can occur late due to a lack of mycology laboratory services. Improvement in laboratory diagnosis of fungal infections is required mycology diagnosis comprising culture, and antifungal sensitivity (AST) is performed in a few reference clinical laboratories only. Most of the hospital laboratories are providing microbiology diagnoses based on microscopy, including gram stains and cultures. For fungal culture, Sabouraud’s agar media is mostly used. Non-culture markers of infection, notably antigen, antibody and molecular tools, are completely lacking. Unfortunately, local fungal data are scarce globally, and most of the cases go undiagnosed or not reported due to a lack of mycologists and insensitive diagnostic methods.

The overall burden of fungal diseases in Oman is unknown and is difficult to quantify because these infections are likely substantially underdiagnosed, and no national public health surveillance exists for any fungal disease. A few studies have reported on the fungal disease in Oman, mostly case reports [4,5,6,7,8,9,10,11,12,13,14,15,16,17,18]. However, excellent efforts were made by scientists from all over the world in collaboration with and led by the Global Action Fund for Fungal Infections (GAFFI) and the Leading International Fungal Education (LIFE) to systematically estimate the global burden of fungal infection. Estimates of the incidence and prevalence of these serious fungal infections are essential in order to identify and highlight public health gaps and inform future priorities. Fungal disease leads to much morbidity and some mortality, and improved recognition, prevention, diagnosis, and treatment are necessary. The variation in the reported incidence of fungal infection is affected by differences in diagnostic criteria and case-finding strategies, as well as by true differences between populations. The objective of this work is to estimate the total burden of serious fungal diseases in Oman to assist in determining public health and research priorities.

## 2. Materials and Methods

### 2.1. Sources of Data

Local epidemiology studies that are fungal disease-based are scarce, whereas other topics are abundant. Globally there is insufficient data on mycology, and Oman is no different. All the available published local literature reporting fungal infection cases were reviewed [4,5,6,7,8,9,10,11,12,13,14,15,16,17,18]. In other situations where there are no local data, a population at risk and their frequencies represented in prevalence or incidence rates were utilized to achieve a valid and precise estimation using the most authentic studies addressing frequencies in different populations at risk. In addition, the search was done in PubMed using the terms “fungi”, “infections”, “Oman”, “Middle East”.

### 2.2. Population

The Omani population demographics data were obtained from the annual health report released in mid of 2018 [1]. The annual health report includes the last and most updated version of the total Omani population, the adult population that used to estimate asthma patients, women under 50 years to highlight recurrent vaginal candidiasis, and other demographic details illustrated in Table 1 and Figure 1. HIV/AIDS statistics were obtained from the annual health report 2018 [1]. The statistics include the total number of patients with AIDS from 1986–2018.

### 2.3. Estimated of Fungal Diseases

Recurrent vulvovaginal candidiasis (RVVC) is defined as four or more episodes per year [19]. No local data were available on RVVC. Hence a systematic review incidence rate-based on a global estimation work was used, where it occurs at an incidence of the rate of 3871/100,000 females [20].

The total number of asthma cases in Omani adults was calculated based on the Delphi technique reported by Sultan Qaboos university study done in 2012, where 7.3% was the prevalence of asthma in adults [21,22,23,24]. Based on prior estimates, severe asthma with fungal sensitization (SAFS) occurs at a prevalence of 3.3% in the asthma population, while allergic bronchopulmonary aspergillosis (ABPA) has three prevalence frequencies; lower, midpoint, and upper range, where the midpoint (2.5%) is proposed to be the best estimation [25], supported by data from Saudi [26]. As severe asthma is probably very uncommon in the expatriate community (mostly migrant workers), we have calculated SAFS only in the indigenous Omani population, unlike ABPA, which is especially common in people from the Indian subcontinent, where we have used the total population.

Cryptococcal meningitis annual incidence in the HIV/AIDS population was taken from local studies; a case study reported by Basu et al. [4] and a retrospective case series by Balkhair et al. [5] that found 16 of 77 cases (22%). It was estimated by Rajasingham et al. [27] that 6% of people living with HIV with low CD4 counts and eligible for ART have cryptococcal antigenemia and 70% develop cryptococcal meningitis. From the same local retrospective study [5], *Pneumocystis* pneumonia (PCP) in the HIV/AIDS population was diagnosed in 18 of 77 (25%) cases. A systematic review and meta-analysis done by Wasserman et al. [28] revealed 15.4% PCP cases complicating HIV people in clinical settings and a higher percentage in inpatients. This figure is more conservative for estimating the total PL-HIV (1566) at risk of PCP. Talaromycosis in Oman has been reported only once [6].

The proportion of oral and esophageal candidiasis was retrieved from the Danish, US and Tanzanian population. Oral and esophageal candidiasis occurs at a frequency of 90%, and 20%, respectively in HIV/AIDS cases with CD4 counts < 200/uL [29,30]. Esophageal candidiasis also occurs in 5% of those on ART [31].

Invasive aspergillosis (IA) annual incidence was estimated in leukemia, transplant, lung cancer and chronic obstructive pulmonary disease (COPD) patients. In two local studies, IA in kidney and allogeneic hematopoietic stem cell transplant recipients identified 2 and 9 cases, respectively [7,8]. IA in hematological disorders was represented as AML (acute myeloid leukemia) and non-AML disorders, which included all lymphoma, non-AML leukemia, multiple myeloma and myelodysplastic syndromes. It was assumed that 10% of AML patients develop IA [32] and an equal number for all non-AML cases. Data on hematological disorders were provided by Cancer incidence in Oman 2015 (Department of Non-Communicable Disease) [9]. In Oman, the crude incidence of all cancers in males was 63.9/100,000, a total of 2940 in 2018. The leukemia prevalence was 7.6% (223 cases), 30% of which are AML (67 cases). The crude incidence of cancers in females is 74.9/100,000 (total 3447), with a leukemia prevalence of 4.73%. Leukemia and AML cases in females were 163 and 49, respectively, following the same approach. Lung cancer as a total figure was released by Globocan database where it was 112 cases in 2018 [33]. Based on Yan et al., IA in lung cancer has a prevalence of 2.6% [34]. COPD prevalence (GOLD stage II-IV) is 3.05%, and if 10.5% are admitted to hospital each year, this yields a total of 13,510 at risk. IA develops in at least 1.3% of these patients [35].

In order to estimate chronic pulmonary aspergillosis (CPA) after tuberculosis (TB), the TB burden in Oman obtained from the World Health Organization (WHO) states 310 TB cases in 2017, of which 67% were pulmonary [36]. TB was the primary underlying disease in 16.7% in low prevalence countries [37]. The calculation of prevalence was as stated previously [32].

Candidemia is conservatively estimated at an incidence rate of 5 per 100,000. To ensure a better estimation, local data on the ratio of cases in different risk groups was used combined with estimation summary values and to account for the lack of local data reporting candidemia in cancer and transplant patients [10,32,38].

There were 14 documented cases of urinary tract infections associated with an outbreak of *Candida auris* in Sohar hospital [10]. Based on Bongomin et al. [32], they estimated an average incidence rate of 29 countries reporting intra-abdominal candidiasis (Candida peritonitis) and revealed a 1.15 incidence in 100,000. Using this approach, a total number of 30 cases will be generated primarily in critical care units.

There is a total of 8 mucormycosis cases obtained from the local literature [11,12,13,14,15], including one case of an expatriate that developed pulmonary mucormycosis with chronic renal failure [16]. We have estimated an annual incidence of 2 per million [32].

Based on local estimation values, fungal keratitis forms 12% of the total number of microbial keratitis [17]. As stated by the annual health report in mid-2018 [19], unspecified keratitis is 2575, 8% documented as caused by fungi. Culture yield was only 65%, and microscopy was not done, so a conservative 12% proportion is fungal in etiology.

Skin infections accounted for 232,362 (bacterial, fungal and viral) of all patients attending 5 dermatology clinics in North Batinah Governorate of Oman; Sohar, Saham, Shinas, Khabura and Suwaiq for 4 years from 1 January 2010 to 31 December 2013 [18].

All calculations were done using either Excel or Word sheets of Microsoft windows. The study does not include any interventions with human subjects; hence ethical authorization was not required.

## 3. Results

### 3.1. Omani population

The Sultanate of Oman is a country that is classified as a high-income country based on the World Bank, with a gross domestic product per capita of $16,415 in 2018 [39]. The climate is characterized by diversity; it is hot and humid in littoral areas, hot and dry in the interior areas with the exception of the Dhofar governorate and mountains with high altitudes that have moderate weather throughout the year. The total population number was obtained from the annual health report released in mid-2018 and considered the most updated and valid reference. Oman has a total population of 4,601,706. The Omani population consisted of 2,579,236 with an additional 2,022,470 are expatriates [1].

### 3.2. HIV/AIDSpPopulation

The population living with HIV were assessed by CD4 and/or viral load parameter in the lab to decide on antiretroviral therapy (ART) eligibility at least once in the last year. There were a total of 3060 HIV cases registered in the annual health report covering all cases from 1986–2018, including 1494 deaths. Two cases were reported in 1985 and one in 1984 [1]. In the last year, 86% were assessed by CD4 and/or viral load parameters to decide ART eligibility. In 2018, there were 1566 people living with HIV infection (PLHIV), of whom 1317 were on ART. Other AIDS/HIV population details are explained in Table 2. It was assumed that those not on ART have a decline in CD4 count to <200/uL over 7 years.

### 3.3. Oral and Esophageal Candidiasis in HIV Population

Candidiasis, in its two forms, emerges as a hallmark of AIDS due to its early manifestation. Oral and esophageal candidiasis is estimated to occur in 16 and 69, respectively, in this population [29,30,31].

### 3.4. Cryptococcal meningitis

Cryptococcal meningitis is a major disease causing a high rate of morbidity in PLHIV. Globally death from cryptococcal meningitis is estimated to be 15% in PLHIV, despite progress in the availability of early antifungal therapies [40]. In Oman, there is probably only one case annually based on a disease prevalence obtained from other countries. There are no data on non-HIV associated cases.

### 3.5. Pneumocystis pneumonia

We estimate 5 cases in the Omani HIV population [28] with 18 cases from the local literature. PCP continues to exist as an AIDS-defining opportunistic illness. It is caused by *Pneumocystis jirovecii*, where humans serve as reservoirs. Inter-human transmission of *P. jirovecii* has been well documented [41,42,43]. There are no data on non-HIV associated cases, but they are likely to be 5 to 10 times as many as in HIV in a diverse patient population [44].

## 4. Respiratory Population

### 4.1. Allergic Bronchopulmonary Aspergillosis and Severe Asthma with Fungals Sensitization

Highlighting the total number of asthma cases is crucial to estimate the burden of ABPA and SAFS. Asthma in Oman has a prevalence of 7.3% based on the Delphi approach [21,22,23,24], a prevalence of 259,275. There are no reported local data on fungal asthma, SAFS or ABPA in Oman. ABPA calculation relied on three prevalence estimations. Lower point, midpoint and upper point were 0.7% (830), 2.5% (2965), and 3.5% (4152), respectively. The midpoint in this study is considered the best speculative estimation [25]. Knowing that the total number of asthma in indigenous Omani adults only is 118,620, the SAFS prevalence estimate is 3910.

### 4.2. Oral Candidiasis Post Inhaled Corticosteroids

About 70% of asthma patients use inhaled corticosteroids (ICS), and as an adverse effect, 6% are estimated to develop oral candidiasis. That would likely give 10,890 oral candidiasis among asthmatic ICS users [45,46].

### 4.3. Chronic Pulmonary Aspergillosis (CPA)

Chronic pulmonary aspergillosis troubling TB survivors was estimated to be 26 cases annually, an incidence rate of 0.32/100,000 [36,37]. Assuming that other pulmonary conditions are more strongly linked to CPA in Oman, we estimate a total prevalence of 156 cases, which assumes an annual mortality of 15%.

### 4.4. Cancer and Immunocompromised Populations

#### 4.4.1. Candidemia

There are probably 145 cases of candidemia in Oman, 69 in critical care units, 126 hospitalized patients with serious underlying disease and 35 cases in unknown categories, respectively. [10,32,38]. Local data were reported by Al Maani et al. [10] and revealed 11 cases. Al-Hatmi et al. recently reported 5 *Candida* cases associated with COVID-19 [38]. As candidemia is detectable in only ~40% of cases of invasive candidiasis, the total annual incidence is at least double the incidence of candidemia.

#### 4.4.2. Invasive Aspergillosis (IA)

Invasive aspergillosis, caused by *Aspergillus* spp., is a significant fungal infection with high mortality. Invasive disease is primarily seen in immunocompromised and critically ill patients. *Aspergillus* species are found in indoor and outdoor environments on surfaces and aerosolized. Once the spores are inhaled, they can germinate into hyphae that invade the mucosa of the respiratory system causing IA. IA cases are on the increase due to increasing numbers of transplant operations [47,48]. In Oman, Royal hospital (RH) started in 2014 performing only hematopoietic stem cell transplantation (HSCT) with an average of 10 HSCT per year. RH did a total of 29 HSCT in early 2016. Out of 29, the indication for HSCT was multiple myeloma in 16, Hodgkin’s lymphoma in 10 and non-Hodgkin’s lymphoma in 3 patients. Sultan Qaboos University Hospital (SQUH) started in 1995 with one single bed high-efficiency particulate air filtered unit (4 cases in 1995), doing both allogeneic and autologous HSCT. A total of 348 HSCT had been done by November 2016. The main indication was malignancy, followed by immunodeficiency and bone marrow failure with an average of 25–28 HSCT per year [49]. IA post kidney transplantation or allogeneic HSCT was 9 and 2, respectively [7,8] (Table 3). AML total cases were calculated based on the crude incidence of cancer cases in Oman [9] and appeared to occur at a frequency of 116 in 2018 in Oman. Using the estimated frequency in 2015 Oman cancer incidence, 10% of the 116 to have IA. Total non-AML cases are 386, 10% of them estimated to have IA [9,32].

#### 4.4.3. Mucormycosis

Mucormycosis is a less frequent fungal infection compared to candidemia and invasive aspergillosis, and we estimated only 8 cases annually. Although low profile, its increasing incidence partly because of diabetes and trauma [50,51], may surpass immunocompromised patients. In Oman, the local literature has recorded eight cases [11,12,13,14,15], including one additional case in an expatriate [16].

### 4.5. Critical Care and Surgery Populations

#### 4.5.1. *Candida* Peritonitis

Intra-abdominal candidiasis, or fungal infection of the peritoneum, is a major emergency encountered in critical care and surgery settings. It has a high rate of morbidity and mortality caused several *Candida* species [52]. There are 35 cases estimated based on an average incidence reporting *Candida* peritonitis in 29 countries.

#### 4.5.2. Urinary Tract Infection (UTI)

Candida UTI or culture-positive urine is a common disease, but here is highlighted as it can worsen or be the marker of outbreaks, as in the situation of *Candida auris*.

### 4.6. Others

#### 4.6.1. Histoplasmosis

Histoplasmosis is a fungal infection of the lung caused by inhalation of *Histoplasma capsulatum* spores. The disease is still neglected and, on many occasions, is misdiagnosed as tuberculosis with a high incident death rate as a consequence of misdiagnosis. Oman most likely has an annual incidence of 3–26 cases. This range reflects all forms of histoplasmosis, not only HIV/AIDS: there are no data available from the local literature [53,54].

#### 4.6.2. Recurrent Vulvovaginal Candidiasis (RVVC)

There are 56,285 women estimated to have RVVC at an incidence rate of 2446/100,000 in the fertile female population of about 1 million [20]. RVVC is a disease of immunocompetent and healthy women usually caused by *Candida albicans* and some other species. RVVC has emerged as a high prevalence problem globally with a profound negative impact on lifestyle quality but is not fatal [55].

#### 4.6.3. Fungal Keratitis

In Oman, there are estimated to be ~300 fungal keratitis cases annually based on local prevalence (12%) provided by Al-Ghafri et al., which is probably conservative [17]. This local prevalence is comparable with the prevalence stated by the global burden of fungal diseases by Brown et al. [56].

#### 4.6.4. Tinea Capitis

Tinea capitis is a particular problem for children who live in the poorest communities. Tinea capitis can lead to kerion development after the scalp gets inflamed and, in turn, secondary bacterial infection, scarring of the scalp and permanent hair loss [25]. Fungal skin infections form a total of 22,303 cases (including tinea capitis; 858 cases) out of 232,362 different skin infections (bacterial, fungal and viral) [18]. Asymptomatic skin colonization (7 cases) from Sohar hospital during *Candida auris* outbreak as reported by Al-Maani et al. [10], yields a total of 22,310 affected.

In Oman, 858 cases of tinea capitis were reported from a study done at the North Batinah Governorate of Oman, i.e., Sohar, Saham, Shinas, Khabura, and Suwaiq. A global estimation revealed that worldwide populations suffer from fungal skin infection with a prevalence of 14.3%. Once applied to the Omani population, this would make 368,831 Omani people affected with fungal skin infections every year [57].

## 5. Summary

Overall, we estimate that 79,520 people suffer from serious fungal disease in Oman, 1.7% of the population (Table 4). This estimate does not include most superficial skin infection, allergic fungal rhinosinusitis and most other infections of the upper airways or otitis externa.

## 6. Discussion

Epidemiological activities of fungal diseases in Oman are restricted in size. Current surveillance papers are few, and no fungal infection is notifiable in the Sultanate of Oman. Thus, this study relied greatly on crude estimates of prevalence and incidence obtained from other populations and countries. This study revealed that 79,520 (1.7%) of the total Omani population suffer from fungal diseases annually, not including fungal skin infections. Hypothetically speaking, the fungal disease burden in the outpatient department is believed to outweigh those in hospital admission. Hair, nails, and fungal skin diseases come at the fourth most frequent issues after headache and dental caries at a global level [58]. Global deaths attributable to fungal disease is estimated to be more than 1.5 million, which is as high as the death rate of tuberculosis (1,500,000) [59]. In Oman, fungal infections with high mortality like candidemia, IA, PCP, and mucormycosis are not as frequent as found in many countries, including Spain [60], but they contribute to poor outcomes, mostly in those with underlying pathology.

Histoplasmosis occurs at an incidence rate of 0.1–1/100,000 in temperate climates, 10–100/100,000 in humid tropics, and >100 cases /100,000 in a high-risk group. In general, histoplasmosis is a very difficult disease to estimate accurately; unfortunately, there is no local data of cases having histoplasmosis, and this the case in many countries worldwide. WHO declared that histoplasmosis is one of the major diseases associated with a high rate of mortality in HIV/AIDS patients [53]. There is one case reported of imported talaromycosis in Oman. Oman is not an endemic area for such infection because the natural reservoirs (Bamboo rats) exist in Southeast Asia. Talaromycosis stands out as an opportunistic illness in AIDS patients. The case reported here is an HIV patient who has a history of recurrent visits to Malaysia, the likely reason for having this type of infection, and emphasizing the importance of travel in importation of unusual fungal infections [6,61,62].

Our study estimations are crude estimates along the lines of calculations done by the Nobel prize winner in physics in 1938; Enrico Fermi provided the order of magnitude estimates. With this in mind, even the large estimation values as in SAFS or ABPA will fall within the one log range, and therefore it can be used as a benchmark to validate future epidemiological studies [63]. We can comfortably say that invasive candidiasis, chronic and invasive aspergillosis and fungal keratitis are in the hundreds, SAFS and ABPA in the thousands and RVVC in the tens of thousands.

Many limitations should be considered for the analysis and interpretation of the results. The study relied largely on rough estimations based on frequencies of diseases from other countries and hence should be regarded with caution. Some patients receive health care in private healthcare settings, which do not feed into any central disease databases. Very few autopsies of deceased patients are done which could uncover concealed cases of lethal mycoses, especially in immunocompromised or critical care patients. The most common missed invasive fungal infection is aspergillosis [64,65,66].

Uncertainty surrounds each estimate, as some may be overestimated or underestimated and even be overlapping. SAFS and ABPA results may overlap because 40% of cases with *Aspergillus* sensitization acquire ABPA, and several cases with ABPA suffer from severe asthma [67]. In 2014, the MoH reported 4500 asthma admissions to the hospital, and many of these will probably have had SAFS. To avoid confusion and invalidate results, we have not estimated chronic obstructive pulmonary diseases (COPD) linked to asthma because, on many occasions, they are clinically misdiagnosed. COPD and asthma cases are usually classified together in National health reports.

There are large untouched data sources of central databases like medical insurance that cannot capture fungal disease accurately because of a lack of sensitive diagnostics and data recording but can capture antifungal treatment. Another source of data gathering could be the receipts from the selling of pharmaceutical products. No research was conducted based on these data sources. Future epidemiological studies are advocated to validate these estimations.

Hospital hygiene level and sanitization systems would play an essential role in preventing or reducing nosocomial diseases in healthcare settings. Periodical environmental screening is needed to ensure clean hospital environments, notably burn units, operating rooms and profoundly immunocompromised patients. Strategic thinking by decision-makers is required to aid in increasing local surveillance activities about the burden of fungal infections and implement appropriate diagnostics. Currently, only culture and microscopy are available in Oman, so the country lacks all the fast and sensitive antigen and molecular tests as well as IgG and IgE serology. Yeast identification can be made by MALDI-TOF in the national reference laboratory.

## 7. Conclusions

With the availability of local data and literature estimates of incidence or prevalence of fungal infection, almost 79,520 (1.7%) of the population in Oman suffer annually from fungal infections, not including fungal skin infections. In light of this, there is substantial uncertainty about these total figures, and hence it is essential to trigger national surveillance campaigns to validate or modify these estimates.

## Figures and Tables

**Figure 1 jof-07-00005-f001:**
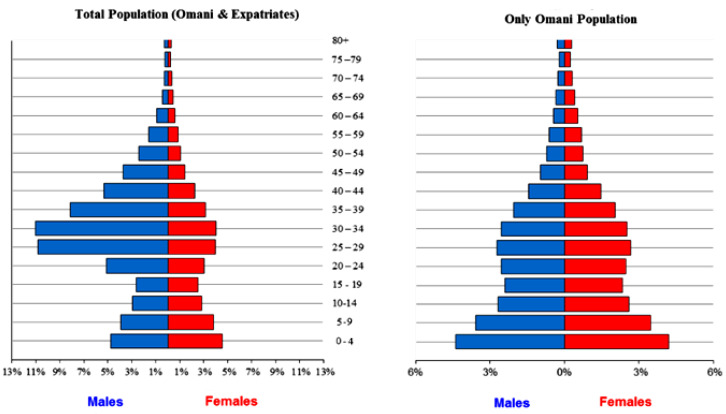
Omani population demographics 2018 [1]. Comparison of the total population (Omani and expatriates) and the Omani population only.

**Table 1 jof-07-00005-t001:** Omani population demographics data from the annual health report 2018.

Population Data	Age (years)	Total	Omani	Expatriate	% Expatriate
Total	All ages	4,601,706	2,579,236	2,022,470	44
Children	0–14	1,050,009	954,317	95,692	9.1
Adults	>15	3,551,697	1,624,919	1,926,778	54.2
Female	>15–<50	938,089	661,348	276,741	29.5
Female	>60	87,666	81,057	6609	7.5

**Table 2 jof-07-00005-t002:** HIV/AIDS population in Oman in 2018.

HIV/AIDS Population	Number, Ratio or Percentage
Total ever reported	3060
Total reported HIV deaths	1494
Current number of PL-HIV	1566
PLHIV male:female	1061:505
% PLHIV assessed	86%
PLHIV on ART	1317
% PLHIV on ART	84%
Likely number < 200/uL CD4	38

PLHIV, the population living with human immunodeficiency virus; ART, antiretroviral treatment.

**Table 3 jof-07-00005-t003:** Invasive aspergillosis in leukemia and transplant populations.

Type	Total	Annual Incidence of IA
AML	116	12
Other hematological disorders	386	39
Kidney Tx	162	9
Allogeneic HSCT Tx	132	2
COPD admissions to hospital	14,737	192
Lung cancer	112	3
Total	15,645	257

IA, invasive aspergillosis; Tx, transplant; HSCT, hematopoietic stem cell transplant; AML, acute myeloid leukemia.

**Table 4 jof-07-00005-t004:** Summary burden of fungal diseases in Oman.

Serious Fungal Infections	Unknown	AIDS/HIV	Respiratory	Cancer/Immunocompromised	Critical Care/Surgery	Total Burden	Rate/100,000
Cryptococcal meningitis		1				1	0.02
PCP		5				5	0.11
Oral candidiasis		16	10,890			10,906	237
Esophageal candidiasis		69				69	1.5
Invasive aspergillosis			192	65		244	5.4
CPA			156			156	3.4
ABPA			6480			6480	141
SAFS			3910			3910	85
Candidemia+	35			126	69	265	5.0
Urinary tract infections					14	14	0.3
Candida peritonitis					35	35	0.75
Mucormycosis				8		8	0.2
Histoplasmosis	3–26					3–26	0.1–1
Recurrent Candida vaginitis (≥4×/year)	56,285					56,285	2446 ^#^
Fungal keratitis	309					309	12
Tinea capitis	858					858	19
Total burden	57,490	92	21,320	163	309	79,520 *	

*—total burden not including fungal skin infections. #—females only.

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
