# Peer review of "Estimated Burden of Fungal Infections in Oman"

_jof, 2020, doi:10.3390/jof7010005_

Round 1

Reviewer 1 Report

The authors outline the global fungal burden in Oman, which is an important healthcare concern.

The burden of fungal diseases is being considered as a health public item all over the world, with many reports from different countries reported in the recent literature.  

However some concerns have to be adressed

The title of the paper state "Estimated burden of serious fungal infections in Oman", while the paper reports on various fungal diseases, including superficial skin infections, not specifically serious

The English need to be enterely revised, with careful attention to sintax and punctuation.

The paper should be better organized, the "material and methods" section contains information repeated in the "results" paragraph. The numeration of subheadings in the "Results" is not clear.

Introduction

line 45-51 the sentence can be shorten, and on line 52 is redundant

Material and methods

This section needs a particular revision of English language, and should be shorten, for example line 97-99 should stay in the Results section, line 08 the paragraph is difficult to read.

Results

line 52-57 Again the paragraph declare information already reported, at line 69 the sentence is not complete. The subheadings need to be re-organizied

Discussion

no major concerns  

Author Response

Reviewer 1:

Comments and Suggestions for Authors

The authors outline the global fungal burden in Oman, which is an important healthcare concern.

The burden of fungal diseases is being considered as a health public item all over the world, with many reports from different countries reported in the recent literature.

Response: Thanks for your nice words and comments in order to improve our manuscript

However some concerns have to be addressed

The title of the paper state "Estimated burden of serious fungal infections in Oman", while the paper reports on various fungal diseases, including superficial skin infections, not specifically serious

Response: We deleted the word ‘serious’

The English need to be entirely revised, with careful attention to sintax and punctuation.

Response: Revised

The paper should be better organized, the "material and methods" section contains information repeated in the "results" paragraph. The numeration of subheadings in the "Results" is not clear.

Response: Corrected

Introduction

Line 45-51 the sentence can be shorten, and on line 52 is redundant

Response: Shorten and corrected

Material and methods

This section needs a particular revision of English language, and should be shorten, for example line 97-99 should stay in the Results section, line 08 the paragraph is difficult to read.

Response: Corrected

Results

Line 52-57 Again the paragraph declare information already reported, at line 69 the sentence is not complete. The subheadings need to be re-organizied

Response: Corrected

Discussion

no major concerns

Reviewer 2 Report

The paper by Al-Hatmi et al. evaluates the burden of fungal infections in the Sultanate of Oman. This first report indicates that apparently 1.7% of people in the Sultanate is affected by serious fungal infections, with a prevalence of 5.4 and 5.0/100,000 for invasive aspergillosis and candidaemia, respectively, while the most commons infection are vaginal candidiasis, followed by allergic respiratory infections. The prevalence data appear lower respect to other countries in the same area (e.g., Kuwait) and are surely underestimated because there are no reporting obligations for fungal infection, and traditional diagnostic methods can hinder the identification of fungi.

Overall, the study confirm that specific surveillance programs and up-to-date diagnostic approaches taking advantage also of molecular/mass spectrometry methods are needed to optimise care and management of fungal infections, especially in some geographic areas.

Author Response

Response: Thanks a lot for your nice words.